# Fine-tuning the efficiency of *para*-hydrogen-induced hyperpolarization by rational *N*-heterocyclic carbene design

Peter J. Rayner [1], Philip Norcott [1], Kate M. Appleby[1], Wissam Iali [1], Richard O. John[1], Sam J. Hart[1], Adrian C. Whitwood [1] & Simon B. Duckett [1]

Iridium *N*-heterocyclic carbene (NHC) complexes catalyse the *para*-hydrogen-induced hyperpolarization process, Signal Amplification by Reversible Exchange (SABRE). This process transfers the latent magnetism of *para*-hydrogen into a substrate, without changing its chemical identity, to dramatically improve its nuclear magnetic resonance (NMR) detectability. By synthesizing and examining over 30 NHC containing complexes, here we rationalize the key characteristics of efficient SABRE catalysis prior to using appropriate catalyst-substrate combinations to quantify the substrate's NMR detectability. These optimizations deliver polarizations of 63% for $^1$H nuclei in methyl 4,6-$d_2$-nicotinate, 25% for $^{13}$C nuclei in a $^{13}C_2$-diphenylpyridazine and 43% for the $^{15}$N nucleus of pyridine-$^{15}$N. These high detectability levels compare favourably with the 0.0005% $^1$H value harnessed by a routine 1.5 T clinical MRI system. As signal strength scales with the square of the number of observations, these low cost innovations offer remarkable improvements in detectability threshold that offer routes to significantly reduce measurement time.

[1] Centre for Hyperpolarisation in Magnetic Resonance, Department of Chemistry, University of York, Heslington YO10 5NY, UK. Correspondence and requests for materials should be addressed to S.B.D. (email: simon.duckett@york.ac.uk)

Hyperpolarization methods that increase the signal strength of magnetic resonance imaging (MRI) are making the in vivo observation of molecular metabolism a clinical reality[1]. This innovation has opened the door to alternative pathways for medical diagnosis. Hence, there is a proven need to broaden the range of materials that can be hyperpolarized whilst simultaneously reducing the cost and complexity of sample delivery[2]. Signal amplification by reversible exchange (SABRE) is a fast growing hyperpolarization process used to overcome the inherent insensitivity of NMR and MRI[3–5]. It derives the associated non-Boltzmann distribution of spin energies from para-hydrogen ($p$-H$_2$) via its reversible binding to a metal catalyst (Fig. 1). Many reported studies use iridium $N$-heterocyclic carbene (NHC) catalysts and polarization is transferred through the resulting scalar coupling network to a ligated substrate molecule[6]. Subsequent ligand dissociation delivers the free substrate where its modified magnetic properties improve detection without changing chemical identity. The binding of fresh $p$-H$_2$ and loss of H$_2$ complete the cycle. The hyperpolarization of nuclei such as $^1$H, $^{13}$C, $^{15}$N has been reported[7–11]. The interest in this method is derived from its relative simplicity and low cost when compared to other techniques[12]. To date [IrCl(COD)(IMes)] (1) (IMes = 1,3-bis(2,4,6-trimethylphenyl)imidazol-2-ylidine, COD = $cis,cis$-1,5-cyclooctadiene) is one of the most effective catalyst precursors in nonaqueous situations[13–15]. A number of theoretical descriptions of SABRE have been reported[16–18] and key spin–spin couplings responsible for polarization transfer quantified[19].

The choice of iridium NHC catalysts reflect their increased efficiency when compared to phosphine systems[3,4,15,20,21]. The ligand identity controls the rate of substrate dissociation and promotes H$_2$/$p$-H$_2$ exchange, essential steps for hyperpolarized substrate formation[14]. These processes have been studied by Exchange Spectroscopy (EXSY) and a pathway involving the intermediate, [Ir(H)$_2$($\eta^2$-H$_2$)(IMes)(sub)$_2$]Cl of Fig. 1, is accepted[15].

Previous studies have shown increasing the steric bulk of the NHC ligand leads to faster substrate dissociation, thus reducing the lifetime of the active catalyst. Hence, SABRE activity is modified[13,14]. The steric bulk of the substrate can also have an effect on the SABRE activity due to inhibited binding[22,23]. Additionally, it has been reported that the rate of magnetic relaxation of the substrate increases in the presence of the SABRE catalyst, with the result that the substrate hyperpolarization decays faster than otherwise expected. This can be mitigated by deuteration of the NHC ligand which increases bound $T_1$ relaxation times such that higher hyperpolarization can be achieved[24–28]. Alternative methods to reduce relaxation include the conversion of SABRE-derived hyperpolarization into longer-lived singlet states and quenching the catalyst by addition of a suitable chelate[29–32].

Previous studies have shown that aromatic NHC ligands are effective mediators of the SABRE process, whereas alkyl NHC's perform poorly[13,14]. This effect could indicate beneficial $\pi$-face interactions[33,34] within the active species[15]. Fully understanding the correlation between electronic and steric NHC properties with SABRE efficacy is therefore essential. This exposes the complexity of the polarization transfer mechanism, where relaxation, scalar coupling and catalyst lifetime all play a role[5,17,18]. More recently, a relationship between the $\pi$-accepting ability parameter (PAAP) and pyridine exchange rates in such complexes was described[35].

A number of analytical NMR methods have now been developed that use SABRE to detect low-concentration analytes and probe diffusion times[36–41]. If the SABRE technology is to become more widely used in industrial and clinical settings, though, it is essential that robust, efficient and predictable indicators of catalytic activity are established. Here we prepare and examine a suite of SABRE catalysts, where their steric and electronic properties are varied systematically. This is achieved by changing the ortho, meta and para substituents on the aryl arms of the NHC ligand and the functionality of the imidazole backbone to access 22 structurally related complexes of the type [IrCl(COD)(NHC)] (1−22) (see Fig. 2). We determine buried volume (%$V_{bur}$)[42–44] and Tolman Electronic Parameter (TEP)[42,45] in conjunction with X-ray crystallography. The SABRE efficacy of these catalysts are quantified for methyl 4,6-$d_2$-nicotinate (A, Fig. 1)[26]. Isotopic labelling of a subset of these NHC ligands results in significantly

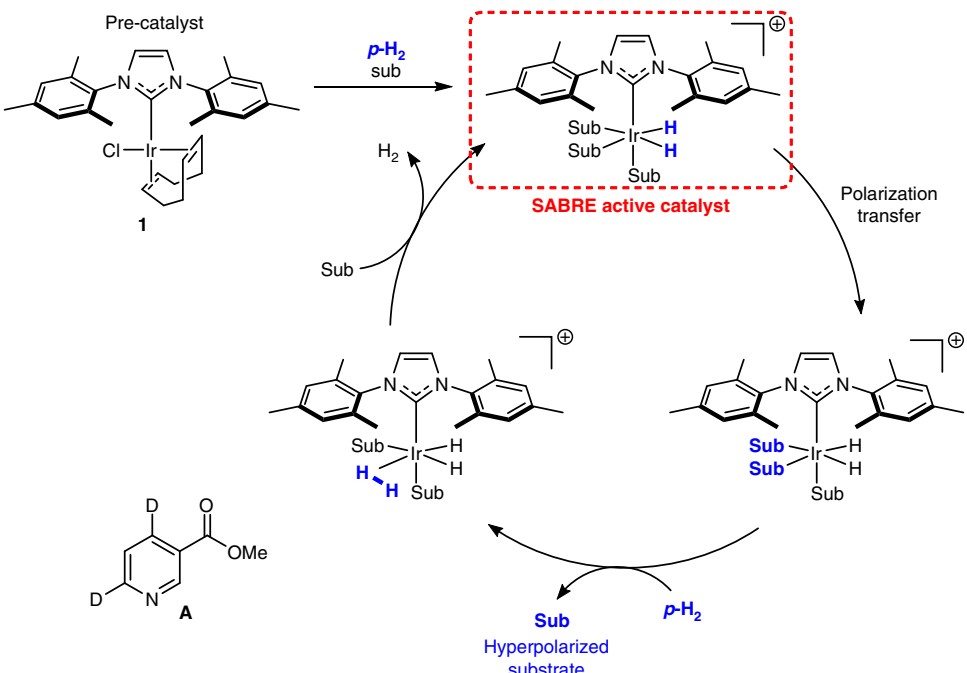

**Fig. 1** Schematic representation of the SABRE catalytic cycle. The reversible binding of substrate (sub) and $p$-H$_2$ leads to the buildup of hyperpolarized substrate in solution which can be detected by NMR or MRI methods

**Fig. 2** NHC identity in the precatalyst [IrCl(COD)(NHC)] **2-22**. Variations in the functional groups in the *ortho* (red), *meta* (orange), *para* (blue) and imidazole (green) positions give access to a diverse range of electronic and steric properties

improved performance. Consequently, we identify optimal SABRE catalyst characteristics that deliver 250% increases in $^1$H polarization levels when compared to those with **1**. Finally, we report on $^{13}$C and $^{15}$N heteronuclear polarization to broaden the scope of the method. For context, hyperpolarization levels achieved reflect over 20,000 (63%), 30,000 (25%) and 125,000-fold (43%) improvements for $^1$H, $^{13}$C and $^{15}$N respectively over the Boltzmann controlled polarizations seen at 9.4 T (400 MHz).

## Results

**Design and synthesis of an *N*-heterocyclic carbene library**. The initial library of 22 iridium NHC catalysts of general formula [IrCl(COD)(NHC)] (**1−22**) were synthesized to encompass a range of substituents in the *ortho*, *meta* and *para* positions of their aryl arms and imidazole backbones. Such variation has previously been shown to lead to increased activity in other catalytic reactions (e.g. transfer hydrogenation[46], borylation[47] and cross-coupling[48]). The catalysts used here are depicted in Fig. 2. Full synthetic procedures and associated characterization data can be found in the Supplementary Methods. In order to define how changing the ligand's functionality in these positions affects their electronic and steric properties, we determined their TEP and %$V_{bur}$ according to standard methods[42,44,45], in addition to determining the X-ray crystal structure for a subset of the precatalysts.

**Electronic effects**. The electron donating properties of the associated NHC ligands were determined as their TEP value, via the corresponding [IrCl(CO)$_2$(NHC)] complexes' IR carbonyl vibrational frequencies[42]. This involved bubbling CO gas through dichloromethane solutions of the corresponding [IrCl(COD)(NHC)] complexes and recrystallization from hexane. The

identity of the carbonyl complexes was confirmed by $^1$H and $^{13}$C NMR spectroscopy as detailed in the Supplementary Methods. Figure 3a reports these TEP values and these data are colour coded to differentiate substituent position effects.

Varying the *ortho*-position substituent from H → Me → Et → iPr (**2 → 3 → 4 → 5**) has little influence on the TEP in accordance with a small inductive change. Similarly, introduction of methyl groups (**6** and **7**) into the *meta*-position results in a minor change in the electron-donating capabilities of the NHC ligand. More substantial electronic effects are evident when the *para*-substituent is varied. Now introduction of electron withdrawing groups, such as halogens (**8−11**), ester (**12**), triflate (**13**) or acetate (**14**) leads to an increase in TEP when compared to **1**. When a phenyl ring (**15**) was located in the *para* position, a small increase in TEP to 2050.3 cm$^{-1}$ is observed. Furthermore, introduction of electron-donating groups such as *tert*-butyl (**16**), methoxy (**17**) or *N,N*-dimethylamino (**18**) all decrease the TEP value. In contrast, changing the substituents on the imidazole backbone significantly influences the electron-donating properties of the carbene ligand. Consequently, when methyl (**19**) or ethyl groups (**20**) are present the carbene becomes significantly more electron donating with TEP values of 2047.2 and 2047.8 cm$^{-1}$ respectively. Chloro (**21**) and bromo (**22**) substituents have the opposite effect, reducing the electron-donating capabilities of the NHC ligand such that their TEP values increase to 2052.8 and 2052.4 cm$^{-1}$ respectively.

**Determining electronic effects from X-ray crystallography**. X-ray crystal structures of nine of the [IrCl(COD)(NHC)] catalysts were solved and their Ir−C$_1$ (where C$_1$ is the carbene carbon) bond lengths analysed to further quantify this effect. All structures are available from the Cambridge Crystallography Data

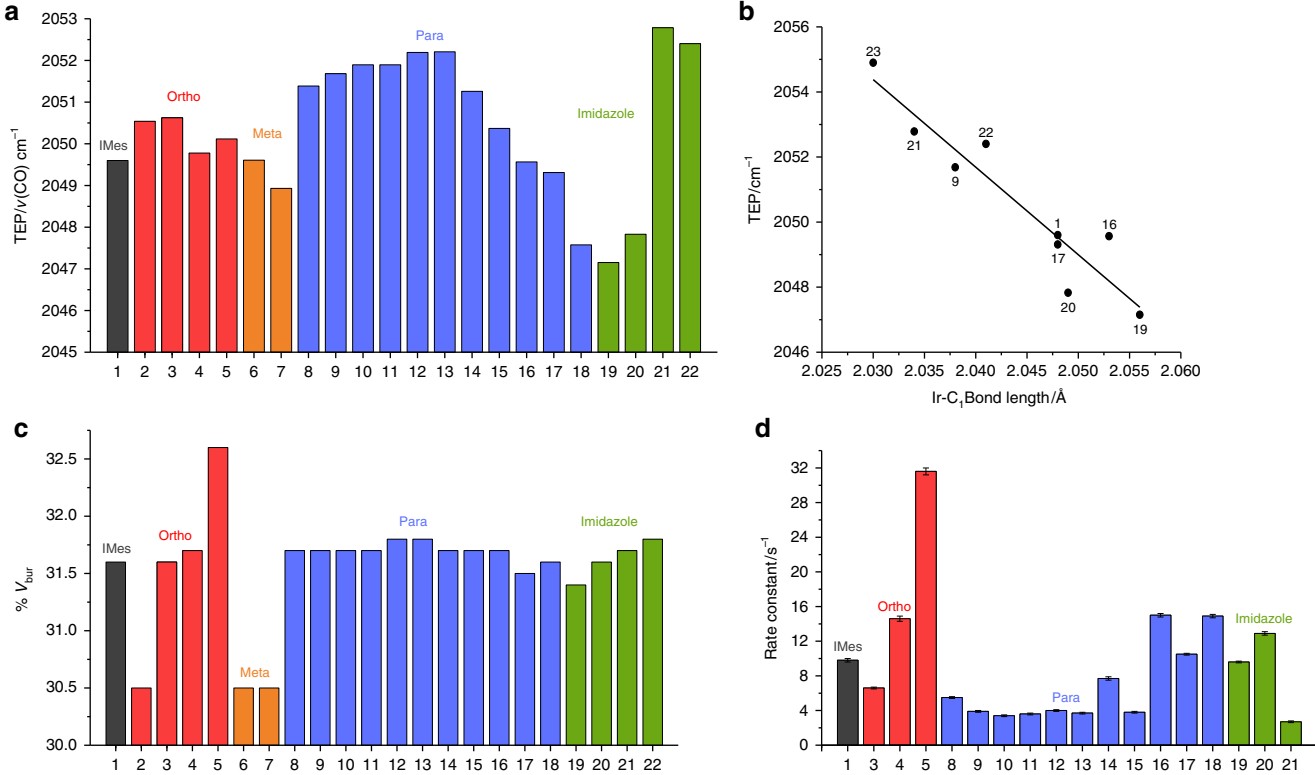

**Fig. 3** Analysis of NHC ligands. **a** Tolman Electronic Parameter (TEP) of the NHC ligand, grouped according to change in the *ortho*, *meta*, *para* and *imidazole* substituent. **b** Relationship between Ir-$C_1$ (where $C_1$ is the carbene centre) bond length and TEP in the corresponding [IrCl(COD)(NHC)]. **c** Buried volume (%$V_{bur}$) value for these catalysts. **d** Rate constants for dissociation of $A_{equ}$ (s$^{-1}$) in [Ir(H)$_2$(NHC)(A)$_3$]Cl (catalysts **2**, **6**, **7** and **22** are omitted due to no active SABRE catalyst formation on exposure to H$_2$)

Centre with full details available in the Supplementary Methods. Interestingly, we found a linear correlation between the Ir−$C_1$ bond length and the TEP with an $R^2$ value of 0.88 (Fig. 3b). Electron-deficient catalysts, such as **21**, exhibit significantly shorter bond lengths than electron-rich variants such as **19**.

**Steric effects**. The steric properties of the NHC were further determined by calculation of their %$V_{bur}$ using the Samb*V*ca Application[44]. Details of the DFT methods employed are available in the Supplementary Methods. Figure 3c shows that the major influence in this steric property results from variation of the *ortho* substituent. When a hydrogen atom is located in the *ortho*-position (**2**) the %$V_{bur}$ reduces to 30.5% when compared to IMes (**1**) or methyl variant (**3**) which both have a value of 31.6%. A slight further increase in steric bulk is achieved by introduction of ethyl substituents (**4**) but this effect is not as significant as that of isopropyl group (**5**, 32.6%). No change in %$V_{bur}$ was observed through the introduction of methyl groups into the *meta* position. Similarly, changes in %$V_{bur}$ caused by modifications to the *para*-substituent are also minimal. Finally, changing the imidazole functionality has a subtle effect on the steric properties of the NHC ligands. For example, a small increase in %$V_{bur}$ is observed through the replacement of protons (**1**) by bromine (**22**).

**Reaction of the NHC catalysts with H$_2$**. A typical active SABRE catalyst has the form [Ir(H)$_2$(IMes)(sub)$_3$]Cl of Fig. 1[24,49]. We therefore screened this catalyst library using the model substrate methyl 4,6-$d_2$-nicotinate[26] (**A**) for this reaction. Hence, a series of samples containing [IrCl(COD)(NHC)] (**1−22**, 5 mM) and methyl 4,6-$d_2$-nicotinate (**A**, 4 eq.) in methanol-$d_4$ were exposed to H$_2$ (3 bar). Analysis by NMR spectroscopy confirmed the formation of [Ir(H)$_2$(NHC)(A)$_3$]Cl as the dominant species in the

majority of cases. Exceptions result for **2**, **6** and **7**, where the *ortho* ring substituent is a proton. For example, **2** led to a new hydride containing complex that yields a $\delta_H$ −12.13 resonance and does not undergo $p$-H$_2$ addition. This is attributed to the formation of a C−H bond activation product based on the reactivity of related phenyl substituted NHC derivatives[50–52]. Furthermore, precatalyst **22**, which bears bromine substituents on the imidazole backbone, does not form a SABRE active catalyst and instead decomposes to a black precipitate on exposure to H$_2$. Therefore, catalysts **2**, **6**, **7** and **22**, which did not form active SABRE catalysts, were removed from the study at this point.

**Rate of substrate dissociation**. The SABRE catalytic activity relates to the rate of loss of $A_{equ}$ in [Ir(H)$_2$(NHC)(A)$_3$]Cl and therefore this parameter was quantified (Fig. 3d)[17]. This behaviour is a consequence of the fact that an optimum complex lifetime is associated with the magnitude of the hydride-hydride and hydride-substrate spin−spin coupling constants it possesses[17]. It can be seen that increasing steric interactions at the *ortho* position significantly increases the rate of substrate dissociation *trans* to the hydride ligand. For instance, the rate constant increases from 6 to 32 s$^{-1}$ by replacing the *ortho* methyl groups (**3**) by isopropyl (**5**). It should be noted that as there are two $A_{equ}$ ligands in [Ir(H)$_2$(NHC)(A)$_3$]Cl, the net rate of $A_{equ}$ loss is twice that reported. The influence of the *para* substituent is less pronounced, with electronic factors outweighing steric effects. Now the plot of TEP versus rate shows a linear trend with an $R^2$ value of 0.90, with the exception of *tert*-butyl catalyst **16** (see Supplementary Figure 42). Catalysts that contain more strongly electron-donating groups than methyl, such as **16−18**, were found to exhibit a higher rate of dissociation than **1**. This can be rationalized by the associated increase in electron density on the metal centre which stabilizes the intermediate

complex and leads to faster iridium−substrate dissociation. Changing the imidazole ring substituent exhibits similar electronic perturbations with electron-poor **21** undergoing slower dissociation ($2.7\,s^{-1}$) than electron-rich **19** and **20** (9.6 and $12.9\,s^{-1}$ respectively). Hence, if SABRE performance is based only upon ligand dissociation kinetics **1**, **17** and **19** should exhibit comparable polarization levels.

**$T_1$ relaxation and SABRE activity.** However, recent experimental and theoretical studies suggest relaxation is also important in controlling catalyst efficiency[17,18,26]. The relaxivity and SABRE activity of the catalyst formed when **1** and **A** combine, which yields 9.8% net polarization in methanol-$d_4$ solution under 3 bar $p$-$H_2$, has been reported[26]. Similar samples containing a 5 mM concentration of the precatalyst (**1**−**21**) and a 20 mM concentration of **A** in 0.6 mL of methanol-$d_4$ are used here to rationalize behaviour. This involved forming $[Ir(H)_2(NHC)(A)_3]$ Cl under 3 bar $p$-$H_2$ and examining the effect of a 10 s polarization transfer time whilst it was located in the 65 G fringe field of the NMR spectrometer. Single-scan [1]H NMR measurements were then made at 9.4 T to assess the associated signal gains for **A** and further measurements conducted to determine the $T_1$ relaxation times via inversion recovery at 9.4 T. The resulting polarization levels, from a minimum of a five observation average, and observed [1]H relaxation times for the two resonances of free **A** are detailed in Fig. 4. As free **A** is in equilibrium with bound **A**, the observed $T_1$ values reflect the ligand exchange rate, and the free and bound relaxation times of **A**[17,26]. In all cases, the concentration and relative excess of **A** was kept constant. The corresponding $T_1$ values at 9.4 T for bound **A** were also determined at 243 K where there is no observable ligand exchange and are presented in Supplementary Fig. 46.

The effect of changing the *ortho*-substituent on both the observed polarization level and the observed $T_1$ values is dramatic. Increasing the steric bulk from Me → Et → [i]Pr (**3** → **4** → **5**) reduces polarization transfer efficiency whilst simultaneously reducing the H-2 $T_1$ relaxation time. As predicted, the overall signal gain decreases by ca. 70% across this series and is proportional to the dissociation rate divided by the relaxation time[17]. Catalyst **5** has the fastest rate of dissociation of **A** and smallest observed polarization levels of just 3.6 and 3.3% for H-2 and H-5 respectively. It is therefore clear from these results that a combination of these two effects is important, although the corresponding catalyst $T_1$ values, determined at 243 K, show little difference. It must therefore be remembered that the free substrate $T_1$ values at 298 K reflect a weighted average of those

of the bound and free substrate and is thus dependent on the ligand exchange rate.

There is also a ca. 40% variation in observed $T_1$ value and polarization level within the series associated with *para*-substituent changes (blue). The introduction of an electron withdrawing halogen increases the polarization level, with fluoro- and chloro-derived catalysts **8** and **9** giving 13.9 and 14.0% net polarization levels respectively relative to **1**. They retain good relaxation times for free **A** because of their slow ligand loss rates. Collectively this change in behaviour delivers an ~40% increase in SABRE efficiency relative to **1**.

Other catalysts containing an electron withdrawing group, triflate (**13**) and acetate (**14**), also showed improvements in SABRE performance relative to **1**. However, while methoxycarbonyl substituted **12** also yields increased net polarization (13.7%), a significant reduction in $T_1$ values of **A** is now seen. In this case, there is also a reduction in the bound H-5 proton $T_1$ value of Ir-**A** at 243 K which would suggest that poor SABRE performance is expected. The observed improvement in SABRE performance by **12** therefore suggests it must actually undergo more efficient polarization transfer than the other materials. This is linked to the magnitude of the $^4J_{HH}$ scalar coupling through which polarization transfer occurs and has previously been suggested to be insensitive to the identity of the NHC[19], albeit for a small range of related examples.

A series of selective 1D-COSY measurements were therefore undertaken at high field to probe this effect through quantification of the corresponding oscillation frequency as detailed in the Supplementary Methods. Compared to **1**, the corresponding value observed for **12** is 10% higher, with a 20% variation being evident across the whole catalyst series. Hence, there is a simple explanation for this behaviour which suggests that rigorous evaluation of this variable is also needed.

Phenyl- (**15**) and *tert*-butyl (**16**)-derived catalysts were also found to reduce the relaxation times of free **A** when compared to **1**. In addition, their $[Ir(H)_2(NHC)(A)_3]Cl$ complexes exhibit fast ligand exchange, similarly large transfer frequencies and short bound $T_1$ ligand values. It is the combination of these effects that results in their poor SABRE performance. Other electron-rich groups, however, perform well with methoxy (**17**) and *N,N*-dimethylamino (**18**) giving 12.3 and 13.7% polarization respectively which are due to their high catalyst $T_1$ values.

The introduction of two methyl groups (**19**) onto the imidazole ring also improves polarization transfer, to give an average polarization level of 13.2% for the two sites of **A** whilst the ethyl derivative (**20**) yields just 4.9% despite extended relaxation times

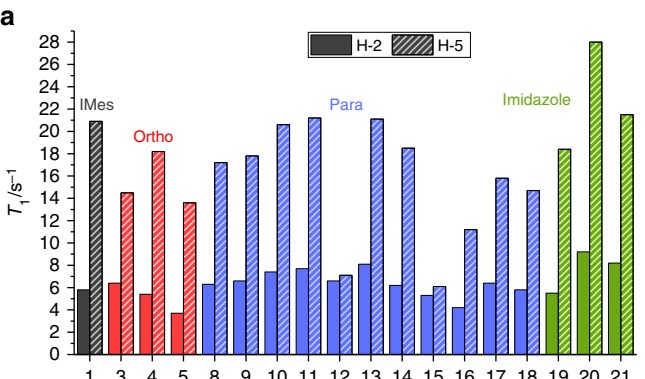
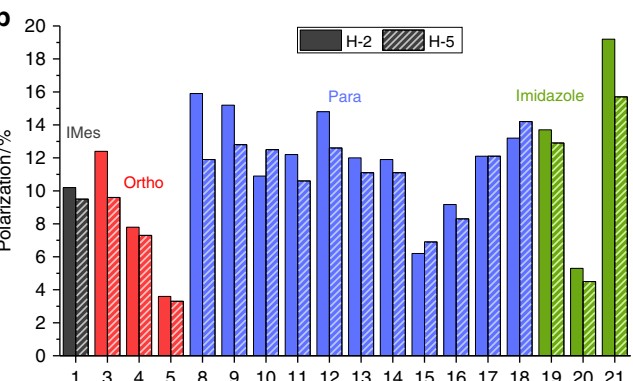

**Fig. 4** Relaxation effects and SABRE performance of catalysts **1**−**21**. **a** $T_1$ relaxation times for H-2 (solid block) and H-5 (dashed) of free **A** (1 eq.) when present with $[Ir(H)_2(NHC)(A)_3]Cl$ (5 mM) and 3 bar $H_2$ in methanol-$d_4$ at 9.4 T. **b** SABRE polarization of the H-2 (solid block) and H-5 (dashed) resonances of **A** in the same sample after SABRE transfer at 65 G under 3 bar $p$-$H_2$. Data are presented as a function of **1**−**21**, and colour coded according to changes in the *ortho* (red), *para* (blue) and *imidazole* (green) substituents

of 8.2 and 28.0 s for H-2 and H-5 of **A** at 298 K. Pleasingly, catalyst **21**, where chloro substituents are introduced on the imidazole ring, results in a 78% increase in net polarization to 17.4% when compared to **1**. There is also a significant increase in the H-2 relaxation time which we suggest is caused by the reduced spin−spin coupling network within the active catalyst. We therefore conclude that these catalyst substituent effects reflect an important but highly complex variable that controls SABRE efficiency. As predicted, it is clear that relaxation within the catalyst itself plays a dominant role in most of these examples[17,26].

**Synthesis and evaluation of deuterated NHC isotopologues.** Deuteration of the carbene ligand can cause an increase in observable SABRE polarization through relaxation time extension of the bound substrate protons whilst simultaneously reducing spin dilution effects[24,26]. Therefore, we synthesized the fully deuterated isotopologues of a subset of the most efficient of these catalysts (Fig. 5). The synthetic procedures are provided in the Supplementary Methods. We measured the associated TEP and % $V_{bur}$ of each of these and found that deuteration had no quantifiable effect on the electronic or steric properties when compared to their protio counterparts. Subsequently, each of the deuterated catalysts were examined and found to improve both the $T_1$ relaxation times and polarization transfer efficiency to **A** when compared to that shown by their protio analogues.

Indeed, $d_{18}$-**3** now achieves 25.5% polarization for H-2 under 3 bar $p$-H$_2$ in methanol-$d_4$ solution. This is over double the level

of its protio counterpart **3**. As predicted, the observed $T_1$ values at 9.4 T for **A** increase from 6.4 and 14.5 s, to 8.7 and 23.3 s, for H-2 and H-5 respectively. A similar effect is observed with $d_{16}$-**9** that now delivers an average polarization of 22.9%. This is 63% larger than that achieved with protio **9** and a 133% increase on **1**. Good levels of polarization transfer are also observed with $d_{16}$-**10** (22.5%) and $d_{18}$-**3** (22.0%) whilst $d_{22}$-**12** and $d_{28}$-**18** deliver less effective increases when compared to their protio variants. Furthermore, when the imidazole ring is functionalized by methyl groups ($d_{22}$-**19**) improved polarization levels compared to **19** result, although the $T_1$ value of H-2 remains just 6.5 s. This suggests that spin−spin interactions between the hydride, bound substrate and methyl group of the imidazole backbone in [Ir (H)$_2$(NHC)(**A**)$_3$]Cl are not innocent. Attempts to reduce this effect by introducing CD$_3$ groups in the imidazole ring proved synthetically unsuccessful due to low levels of isotope retention. However, chloro derivative $d_{22}$-**21** was readily prepared and yielded the largest polarization level for both H-2 (26.9%) and H-5 (21.8%) in conjunction with the longest free **A** proton $T_1$ relaxation times, which are 13.8 and 31.7 s for H-2 and H-5 respectively.

The most effective catalysts for the hyperpolarization of **A** for this series of mono-substituent variations have chloro groups on the imidazole ring or *para*-chloro aryl substituents. When these modifications are combined to create $d_{16}$-**23** the revised ligand's TEP value is 2054.9 cm$^{-1}$ and its % $V_{bur}$ is 31.8% (Fig. 5). It is therefore the most electron deficient of these NHCs and the observed dissociation rate constant for **A** is just 1.4 s$^{-1}$ in the

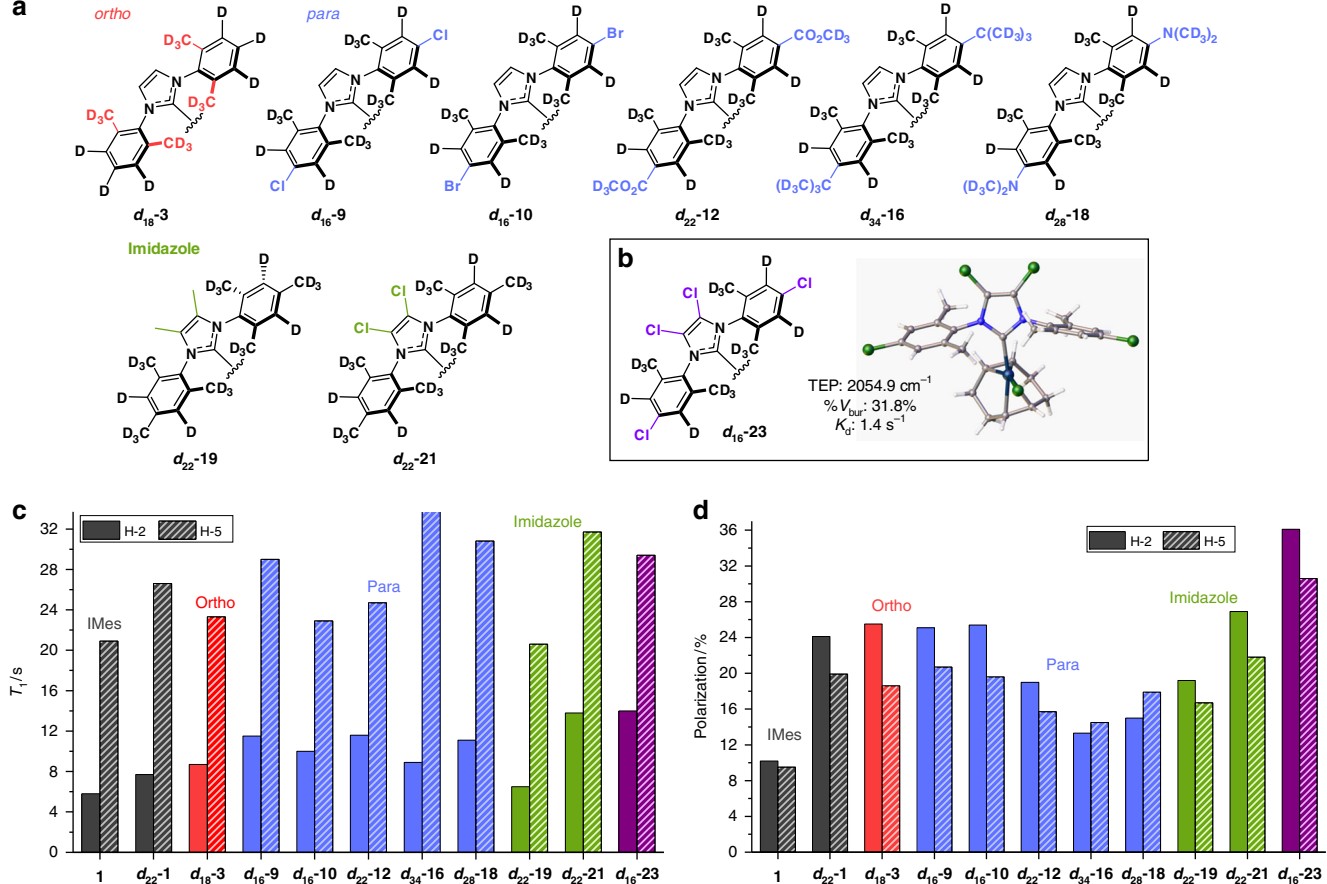

**Fig. 5** Deuterated NHC isotopologues. **a** Structures of isotopically labelled catalysts. **b** Structure of tetrachloro derivative $d_{16}$-**23** and its TEP, % $V_{bur}$ and $K_d$ data and X-ray crystallography structure which has been deposited in the CCDC: 1820380. **c** $T_1$ relaxation values at 9.4 T for H-2 (solid block) and H-5 (dashed) of **A**. **d** Corresponding SABRE polarization levels achieved

corresponding catalyst at 298 K. This process exemplifies the route to second-generation highly optimized catalysts for SABRE.

A sample containing precatalyst $d_{16}$-23 (5 mM) and **A** (4 eq.) in methanol-$d_4$ was exposed to 3 bar $p$-H$_2$ and polarization transfer undertaken at 65 G. Under these conditions, polarizations of 36.1 and 30.6% for H-2 and H-5 respectively were achieved in conjunction with a 140% extension in the $T_1$ relaxation time of H-2 to 14.0 s over that seen with **1**. When a 5 bar pressure of $p$-H$_2$ was employed, the H-2 polarization level increased to 40.7%[26]. In a final modification, when 3 eq. of the fully deuterated cosubstrate, methyl 2,4,5,6-$d_4$-nicotinate[26], is introduced in combination with **A** (1 eq.), 63% polarization for H-2 is achieved. Thus, $d_{16}$-23 is the most effective in this series of catalysts for the hyperpolarization of **A**. For context, this polarization value would be expected to result in a 130,000-fold gain in signal strength if **A** were now to be observed at the common clinical MRI field of 1.5 T. It is clear from these data that utilization of a $^2$H-labelled catalyst changes the dominant effect that $T_1$ relaxation plays on SABRE performance, that is seen with their $^1$H-labelled counterparts, so that catalyst lifetime must now match the $^4J_{HH}$ coupling or polarization transfer frequency[17,18].

**Expanding the substrate range**. Having optimized SABRE for methyl 4,6-$d_2$-nicotinate (**A**), a wider set of substrates were tested to probe the importance of such variations more generally. This involved screening a subset of these catalysts against the substrates shown in Fig. 6. These substrates encompass electron-poor (3-nitropyridine (**B**), 3-trifluoromethylpyridine (**C**)) and electron-rich heteroaromatic (3-picoline (**D**), 3-methoxypyridine (**E**), 3-(N,N-dimethylamino)pyridine (**F**)) examples. Figure 6 reveals the best performing catalyst, and that of its $^2$H-labelled counterpart for each substrate, in addition to the performance of **1** for comparison purposes. The quoted polarization levels are the average value that result across the four proton sites of the specified substrate, a full breakdown of these polarization data and associated effective relaxation times can be found in Supplementary Tables 4-13.

It can be seen that the best performing catalyst for 3-nitropyridine (**B**) is the *para*-dimethylamino substituted derivative **18**. It achieves 3.8% polarization of **B**, compared to 1.8% using **1**. Further improvements were gained by using $d_{28}$-18 which gave 5.1% average polarization. For 3-trifluoromethylpyidine (**C**) catalyst **21**, with chloro substituents on the imidazole ring, gave

the highest average polarization level of 2.7% which reflects a 67% increase in performance over **1**. Its deuterated derivative, $d_{22}$-21 delivers 8.1% polarization. A fourfold improvement in average polarization level of 3-picoline (**D**) was observed with ester derived **12** and its isotopologue $d_{22}$-12, when compared to **1** as average polarization levels of 5.4 and 5.8% respectively are achieved. For both 3-methoxypyridine (**E**) and 3-(N,N-dimethylamino)pyridine (**F**), *para*-chloro containing **9** was optimal, leading to 5.6 and 3.6% polarization levels respectively. Now $d_{16}$-9 gave substantial further improvements reaching 10.8 and 8.1% values respectively. These values are now limited by the substrate $T_1$ values and it might be expected their $^2$H-labelling or employment of higher $p$-H$_2$ pressures would again result in higher polarization levels[26]. These data therefore confirm the importance of such systematic studies in conjunction with $^2$H-labelling if SABRE is to be used to optimally detect a given substrate.

**Application to $^{15}$N and $^{13}$C nuclei**. Our final aim is to demonstrate that the same approach can be used to give improvements in heteronuclear signal detection. Theis et al. have previously shown that direct polarization transfer to pyridine-$^{15}$N can be achieved in microtesla fields through SABRE-SHEATH (SABRE in SHield Enables Alignment Transfer to Heteronuclei)[11,29,53]. A series of samples containing pyridine-$^{15}$N (25 mM) and [IrCl(COD)(NHC)] (2.5 mM) were therefore examined in methanol-$d_4$ under 3 bar $p$-H$_2$ after polarization transfer in a μ-metal shield (ca. 350-fold shielding, see Supplementary Methods). Catalyst **1** yielded 7.1% $^{15}$N-polarization under these conditions as quantified by comparison to a single-scan, thermally equilibrated 5.0 M solution of $^{15}$NH$_4$Cl according to standard methods[54] (Fig. 7). Screening revealed that the optimum catalyst was **21**, with chloro imidazole ring substituents, which improved this to 11.0%. To rationalize these results, we quantified the rate of pyridine loss as 4.8 s$^{-1}$ and is significantly slower than that of [Ir(H)$_2$(IMes)(Py)$_3$]Cl which is 11.2 s$^{-1}$ [14]. The enhancement could be improved by use of the deuterated isotopologue, $d_{22}$-21 to 15.5%. This change is reflected in the ~20% extension in magnetic state lifetime at 9.4 T which is now 28.7 s rather than 24.2 s for systems based on $d_{22}$-21 and **1** respectively. The polarization of pyridine-$^{15}$N can be improved further to 42.3% when $d_{22}$-21 (2.5 mM) is employed with pyridine-$^{15}$N (6.25 mM) and pyridine-$d_5$ (18.75 mM) in methanol-$d_4$ under 5 bar $p$-H$_2$. The pyridine-$d_5$ co-ligand achieves this effect by reducing spin dilution whilst extending further the lifetime of the hyperpolarized state to 38.0 s.

We also improve the SABRE polarization of the $^{13}$C nuclei in doubly labelled $^{13}$C$_2$-diphenylpyridazine (**G**), that has been shown to sustain a singlet state with a lifetime of ca. 2 min[32], though other substrates have been shown to be amenable to $^{13}$C SABRE[25,55–57]. Polarization transfer using **1** (5 mM) and **G** (20 mM) in methanol-$d_4$ at ~0.5 G resulted in a 2.6% polarization level (Fig. 7). This improved to 5.6% with *tert*-butyl-substituted catalyst **16**. We attribute this change to the increased rate of substrate dissociation in the corresponding SABRE catalysts which are 0.20 and 0.46 s$^{-1}$ for **1** and **16** respectively. A change in the rate of haptotropic shift is also observed[49] and full details can be found in Supplementary Table 17. Deuterated isotopologue $d_{34}$-16 improved the observed polarization level to 8.2% under analogous conditions. By reducing the concentration of $d_{34}$-16 to 2.5 mM and utilizing a 1:3 ratio of **G** and its fully deuterated isotopologue, 4,5-di(phenyl-$d_5$)-3,6-$d_2$-pyridazine, a 25.0% polarization level was achieved with 5 bar $p$-H$_2$. This level of hyperpolarization is commensurate with that created by hyperpolarization techniques that have already been successfully used for in vivo biomedical imaging[58–62].

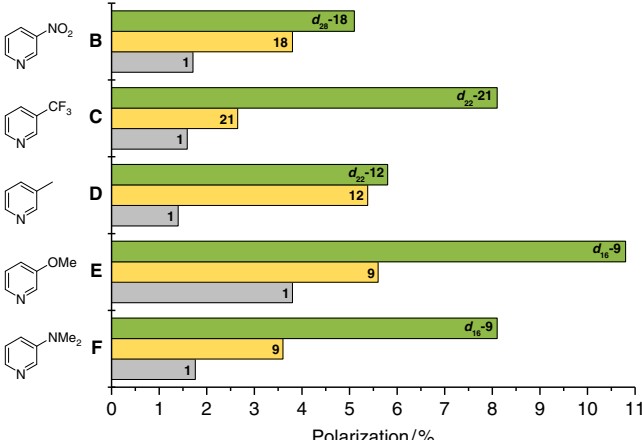

**Fig. 6** Expanding the substrate scope. Polarization transfer performance for the indicated precatalyst [IrCl(COD)(NHC)] (5 mM) and substrate **B**–**F** (4 eq.) in methanol-$d_4$ under 3 bar $p$-H$_2$ as protio (yellow), deuterio (green) and **1** (grey)

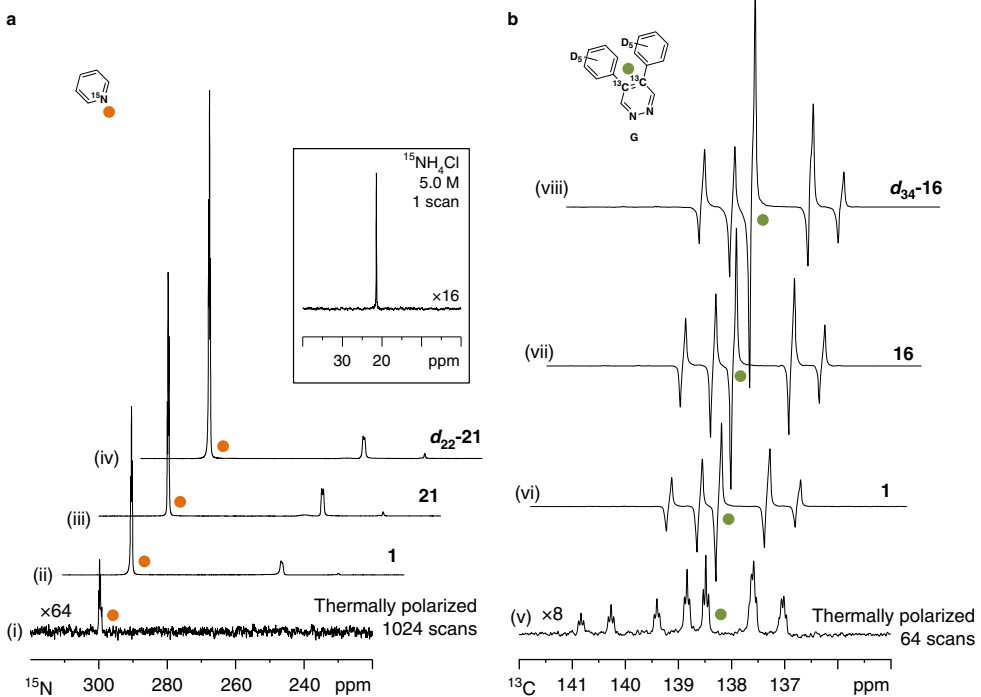

**Fig. 7** Heteronuclear SABRE hyperpolarization. **a** $^{15}N$ NMR spectra of pyridine-$^{15}N$ (25 mM) achieved with indicated [IrCl(COD)(NHC)] (2.5 mM) precatalyst using the conditions: (i) thermal polarization recorded with 1024 scans (×64 vertical expansion); (ii) single-scan SABRE polarization using **1**; (iii) single-scan SABRE polarization using **21**; (iv) single-scan SABRE polarization using **$d_{22}$-21**. Peak at ca. 254 ppm is attributed to equatorially bound pyridine-$^{15}N$ in the active catalyst[11]. Inset: Single-scan $^{15}N$ NMR spectrum of $^{15}NH_4Cl$ (5.0 M) in $D_2O$. **b** $^{13}C$ NMR spectra of **G** (20 mM) using indicated [IrCl(COD)(NHC)] (5 mM) and the conditions: (v) thermal polarization visible after 64 scans (×8 vertical expansion); (vi) single-scan SABRE polarization using **1**; (vii) single-scan SABRE polarization using **16**, (vii) single-scan SABRE polarization using **$d_{34}$-16**; peaks are assigned as previously reported[32]. Measurements used 3 bar $p$-$H_2$

## Discussion

In summary, we have improved the levels of SABRE hyperpolarization that can be created in the $^1H$, $^{13}C$ and $^{15}N$ nuclei of a range of substrates by utilization of an optimized catalyst. This involved establishing that SABRE efficiency was linked to the identity of the NHC in the catalyst while detailing how it influences magnetic relaxation, the rate of substrate dissociation and the size of the $^4J_{HH}$ coupling within the catalyst[5]. This required the preparation of a series of SABRE catalysts whose NHC ligands were specifically synthesized to encompass a wide range of steric and electronic properties. Collectively, these data illustrate a robust route to improve final polarization levels across a wide range of substrates and nuclei. In particular, modifying the *ortho*, *meta* and *para* functionality of a series of aryl-substituted NHC ligands, in conjunction with changing the substituents present on the imidazole ring is found to have a substantial effect on active catalyst lifetime, delivered polarization level and $T_1$ relaxation time.

The results show how changes in steric bulk at the *ortho* position can be used to dramatically increase the rates of substrate dissociation and work to improve SABRE polarization levels for strong-binding substrates. The associated *ortho* substituent effects on the electronic properties of these ligands are small because they induce only small inductive changes. Additionally, we found that when the *ortho* position substituent is a proton, SABRE is quenched due to catalyst instability.

Modifications at the *para*-position led to significant electronic differences which change the substrate dissociation rate and can be used to improve SABRE performance. The most substantial electronic perturbations were caused by changes to the imidazole ring, where the introduction of chloro substituents significantly

slowed ligand dissociation whereas methyl group addition had the opposite effect.

$^2H$-isotopic labelling of the NHC ligand lead to improved SABRE efficacy as a result of extending the bound substrates' polarization lifetimes in all cases. Fine-tuning of these properties yielded polarization levels that were up to six times larger than those achieved by the reference precatalyst [IrCl(COD)(IMes)] (**1**) in conjunction with the test substrate methyl 4,6-$d_2$-nicotinate (**A**) and the tetrachloro substituted catalyst **$d_{16}$-23**. This catalyst proved to be the most electron deficient of those studied and this slowed the rate of dissociation of **A**. Slower exchange maintains the spin−spin network responsible for transfer of the $p$-$H_2$-derived polarization for a time period which is commensurate with the small couplings implicated[18]. The optimum rate of substrate dissociation has previously been predicted to be lower than those commonly observed[17] and is confirmed by these data. Hence, the effects of magnetic relaxation during the lifetime of the active catalyst are important because it is destructive to the SABRE process[17,18,26]. This effect is decreased here by use of $^2H$-labelling in the NHC ligand and accounts for the superior performance of **$d_{16}$-23**. We have demonstrated that the spin−spin couplings vary within these catalysts. However, as the final level of polarization is also associated with the rate of substrate dissociation and spin relaxation[18], optimization will be needed for each substrate.

A range of substrates that encompass both electron-rich and deficient heteroaromatics were also screened. SABRE-delivered polarization levels vary according to the steric and electronic properties of the substrate and catalyst. When they are matched, an optimum catalyst is identified as detailed in Fig. 6. Similar behaviour was observed when $^{15}N$ or $^{13}C$ polarization levels were

investigated. As such, 42.3% $^{15}$N polarization (>125,000-fold signal gain) was achieved for pyridine-$^{15}$N when using $\boldsymbol{d_{22}}$-$\boldsymbol{21}$ as the precatalyst in conjunction with pyridine-$d_5$ and 5 bar $p$-H$_2$. Additionally, $^{13}$C polarization levels of up to 25.0% were achieved for a $^{13}$C$_2$-diphenylpyridazine (**G**) with $\boldsymbol{d_{34}}$-$\boldsymbol{16}$. Catalyst selection is therefore a critical factor in delivering optimal nuclear polarization levels in all of these substrates. Thus, we predict that substrates such as the well-studied metronidazole, which has already yielded excellent $^{15}$N polarization levels of >34% using **1**, could be amenable to further improvement using the catalysts reported here[63,64]. As this is an antibiotic and hypoxia probe, improved catalyst selection could expedite its use in in vivo monitoring. Additionally, whilst the highest levels of polarization reported in this study are obtained at low substrate concentration, this is not necessarily a limit for biomedical applications of SABRE as improved access of the active catalyst to $p$-H$_2$ may lead to improved polarization at higher concentrations and even neat liquids[22].

We believe therefore that the suite of catalysts and the trends illustrated here will contribute to the goal of employing hyperpolarized contrast agents in vivo via SABRE. Ultimately, a bolus may be created in a biocompatible solvent mixture by polarization directly in aqueous media[65–67] whilst utilizing a catalyst sequestering technique[64,68,69], or via biphasic[70] or heterogeneous catalysis[71,72]. Additionally, we expect these developments to be directly applicable to in-high-field methods, such as LIGHT-SABRE[73], and they will also influence the polarization outcome of SABRE-RELAY[74]. As NHC ligands are widely used in catalytic transformations, we expect these data also to impact on other reaction outcomes[75–77].

## Methods

**Materials**. All compounds and solvents were purchased from Sigma-Aldrich, Fluorochem or Alfa-Aesar and used as supplied unless otherwise stated. For detailed synthetic procedures and characterization data for the compounds synthesized in this manuscript, see Supplementary Methods. For $^1$H, $^{13}$C NMR spectra of the compounds synthesized in this manuscript, see Supplementary Figures.

**Sample preparation and SABRE method**. A 5 mm J. Young's tap NMR tube containing a 5 mM (unless otherwise stated) solution of [IrCl(COD)(NHC)] and substrate (4 eq.) in methanol-$d_4$ (0.6 mL) was degassed prior to the introduction of $p$-H$_2$ (3 bar unless otherwise stated). Samples were then shaken for 10 s in the specified polarization transfer field before being rapidly transported into the magnet for subsequent interrogation by NMR spectroscopy. For $^1$H, the typical polarization transfer field was 65 G, for $^{13}$C it was 0.5 G and for $^{15}$N the sample was shaken inside a μ-metal shield with ca. 350-fold shielding.

**Determination of Tolman Electronic Parameters**. CO$_{(g)}$ was bubbled through a solution of [IrCl(COD)(NHC)] in CH$_2$Cl$_2$ for 2 min. The resulting solution was concentrated under reduced pressure. The resulting crude solid was triturated with hexane to give the desired [IrCl(CO)$_2$(NHC)] complex and characterized by $^1$H and $^{13}$C NMR data which is detailed in full in the Supplementary Methods. The carbonyl frequencies of the [IrCl(CO)$_2$(NHC)] complexes were measured in a CH$_2$Cl$_2$ solution using a Bruker Tensor 37 FTIR spectrometer. Full IR data can be found in Supplementary Table 1. The Tolman Electronic Parameter was then calculated from the average carbonyl frequencies using the following equation:

$$\text{TEP}\left(\text{cm}^{-1}\right) = 0.847[\nu_{CO}(\text{average})] + 336.$$

**Determination of buried volume**. All density function calculations were undertaken at the GGA level with the Gaussian09 set of programs. The BP86 functional was used for the optimizations and were carried out in the gas phase. The basis sets with polarization functions (TZVP keyword in Gaussian09) were used for H, C, N, Cl, O, F and S atoms. For the Ir atom, the SDD basis set and associated ECP was used. The geometries obtained were then used to obtain the % buried volume parameters with the SambVca 2.0 website (https://www.molnac.unisa.it/OMtools/sambvca2.0/index.html). The calculations performed here are at the same level of theory used in the creation of this web tool. Further details are available in the Supplementary Methods and full Cartesian coordinates are available from the York Data Catalogue.

## Data availability

For the experimental procedures and NMR analysis of the compounds in this article, see Supplementary Methods and Supplementary Figures in the Supplementary Information File. All data created during this research are available by request from the York Research Database (https://pure.york.ac.uk/portal/en/datasets/search.html). The X-ray crystallographic coordinates for the structures reported in this article have been deposited at the Cambridge Crystallographic Data Centre (CCDC) and can be obtained free of charge via www.ccdc.cam.ac.uk/data_request/cif (accession numbers 1820372–1820380 and 1823650).

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

## Acknowledgements

This work was supported by The Wellcome Trust (Grants 092506 and 098335).

## Author contributions

P.J.R., P.N. and K.M.A. synthesized and characterized the catalysts and ligands. P.J.R. and P.N. determined the TEP values. R.O.J. conducted the DFT and buried volume analysis. P.J.R., P.N. and K.M.A. collected and interpreted the NMR data. W.I., S.J.H. and A.C.W. collected and analysed the X-ray crystallography. P.J.R., P.N. and S.B.D. defined the study and wrote the manuscript. All authors have given approval to the final version of the manuscript.

## Additional information

**Competing interests:** The authors declare no competing interests.

