## [Peer Review File · Nature Communications]

Reviewer #1 (Remarks to the Author):

This is a rather exhaustive study in which the authors attempt a systematic evaluation and rationalisation of several factors that can potentially affect the levels of hyperpolarisation achievable in SABRE. It will be of interest and of practical use for those working in the field of SABRE hyperpolarisation. However, in my opinion this work is not suitable for publication in Nature Communications for several reasons.

The authors tend to overemphasize the significance of their results. While the word 'dramatic' and its derivatives are used a dozen times throughout the manuscript (including the title), an additional signal enhancement of 2-3-fold in NMR, can hardly be called dramatic compared to the enhancement of several orders of magnitude that has been achieved in previous studies. The achieved improvement in SABRE efficiency is certainly useful, but is rather an incremental achievement rather than a major one.

From the fundamental point of view, there is not much novelty in this work either. The electronic and steric properties of NHC ligands and their effects on catalyst behaviour have been addressed previously, both in the general catalytic context, and in the context of parahydrogen-based NMR signal enhancement optimisation. The prolongation of polarisation lifetime and an increase in polarisation levels upon deuteration of ligands and substrates has also been reported before. In particular, all this is mentioned in the introduction which also provides relevant references.

The phrase "propagating scalar couplings" is most likely slang rather than an actual term and should be avoided throughout the manuscript

The manuscript needs careful proofreading as it contains a number of typos and incorrect sentences. The examples are:

where it's modified magnetic properties improve detection...

we show demonstrate the utility of these catalysts...

by replacing then ortho methyl groups...

with electron-poor 21 undergoing slowing dissociation (2.7 s⁻¹) than electron-rich 19 and 20...

the concentrations and relative excess of A was kept constant.

Supporting Information (Fig. SXX).

we rationalize by reduced the spin-spin coupling network within the active catalyst.

Our final aim in this work is to demonstrate...

A series of samples containing...

and its fully deuterated isotopologue...

has previously been predicted to be than those commonly reported...

Reviewer #2 (Remarks to the Author):

This manuscript describes a significant advance in the field of SABRE hyperpolarization, and would be of interest to all working in that field of hyperpolarization. The work is aimed at a more rational catalyst design. The study is thoroughly performed with painstaking 138 pages of supporting information containing 122 supporting figures. I have never seen such a long Supporting Information file. This is a compliment, because the contained information will surely be appreciated by those working in this field and developing novel imaging strategies (clearly outlined in the introduction). I highly recommend this manuscript for publication in this top journal. The manuscript is very well written, and I have only a few minor suggestions.

1) When discussing steric hindrance effect, the authors should cite previous work (R. V. Shchepin, M. L. Truong, T. Theis, A. M. Coffey, F. Shi, K. W. Waddell, W. S. Warren, B. M. Goodson, E. Y. Chekmenev, *J. Phys. Chem. Lett.* 2015, 6, 1961-1967), where the importance of this effect was realized, and elegantly solved in this paper.

2) Most of the record polarization levels reported here are obtained by substrate/catalyst dilution. This may be perceived by some reader as the limitation in the context of biomedical application, where both concentration and polarization are important factors governing the attainable SNR in MRSI studies. I suggest adding the following text (somewhere): "The experiments performed here reported on the record level of polarization values at relatively low substrate concentration to demonstrate the feasibility of high polarization levels. This is not necessarily a limitation for biomedical applications, because the improved access of substrate/catalyst to parahydrogen may enable such high polarization levels at significantly higher concentrations." And cite the above-mentioned paper. This can be done in heteronuclear section of the manuscript.

3) In the sentence: "While metronidazole has already been shown to yield >20% ¹⁵N polarization using 1 this result should therefore be amenable to further improvement using the catalysts reported here.⁵⁷" The comparison is not necessarily fair, b/c Ref. 57 reported this polarization level at 150 mM substrate concentration. Moreover, more recent work reported on >34% polarization at 20 mM concentration: B. E. Kidd, J. L. Gesiorski, M. E. Gemeinhardt, R. V.

Shchepin, K. V. Kovtunov, I. V. Koptug, E. Y. Chekmenev, B. M. Goodson, J. Phys. Chem. C 2018, DOI 10.1021/acs.jpcc.1028b05758. This new paper should be additionally cited by the authors.

4) The authors may also want to cite the above reference in the next paragraph, where they discuss recently developed tools for translation of SABRE in the realm of biomedical imaging.

5) The sentence: "This level of hyperpolarization is commensurate with that created by DNP and successfully used for in vivo study.⁵⁶" should be revised. The choice of DNP reference is subjective. Surely, contrast agents relying on parahydrogen have been demonstrated too. One way to remedy is to revise to "This level of hyperpolarization is commensurate with that created by the hyperpolarization techniques, which are already employed for in vivo biomedical imaging." And cite relevant papers.

6) The authors discuss relaxation, but surely this is a relaxation in mT or high-field regimes. The relaxation at micro-Tesla magnetic fields is very much different: R. V. Shchepin, L. Jaigirdar, E. Y. Chekmenev, J. Phys. Chem. C 2018, 122, 4984–4996. Since the micro-Tesla fields were employed here too, the authors may want to write a clarifying statement.

Reviewer #3 (Remarks to the Author):

In their manuscript, Rayner et al. tested the performance of more than 30 novel N-heterocyclic carbene (NHC) complexes (catalysts) in the hyperpolarization process Signal Amplification By Reversible Exchange (SABRE). The key characteristics of the catalysts were rationalized (i.e., electronic, steric properties and nuclear spin relaxation) and the most important parameters affecting the hyperpolarization efficiency were identified. For the first time, the rational design of the SABRE catalysts was driven by the previously developed models and the data obtained confirms the existing theory. One of the most important conclusions is that catalyst optimization (fine-tuning) is required for each substrate.

This is an excellent paper which deserves to be published in Nature Communications after minor comments are addressed, see below:

1. Title of the paper should be modified. Current title does not explain what the paper is going to be about and looks more like a general description of the SABRE process. Moreover, in the last several years Duckett and co-authors published a number of papers having very similar titles (for example, see Rayner et al., Proc. Nat. Acad. Sci., 2017; Lloyd et al., Catal. Sci. Technol., 2014; Adams et al., Science, 2009). I suggest the following title: "Fine-tuning the efficiency of parahydrogen-induced hyperpolarization experiments by rational catalyst ligand modifications" or something similar which highlights the actual content of the paper.

2. Abstract. Avoid the word “non-hydrogenative” when describing SABRE process. While it is generally true that hydrogenation reaction is not required for the observation of the SABRE phenomenon, several studies (Emondts et al. *ChemPhysChem*, 2018; Barskiy et al., *JACS*, 2014; Glöggler et al., 2011) have shown the presence of the chemical exchange accompanying the polarization transfer process. Therefore, rigorously, it is incorrect to call SABRE a non-hydrogenative process.
3. Page 3, line 86: remove “show” or “demonstrate”.
4. Figure 3d. Where is the data for meta-substituents? Was not it measured? If yes, why? It should be specified in the text.
5. Page 8. Lines 162-166. Formation of a new complex showing the hydride resonance sounds interesting. Could authors provide more details about the proposed structure (e.g., by putting it in supporting information)? Why does not precatalyst 22 form the active catalyst form? It is only different from other catalysts by the subtle change of the ligand. What is the authors’ explanation? More discussion is necessary even if it is just a speculation.
6. Page 9, line 202: Fig. SXX – please specify.
7. Page 14, lines 349-351. Are the reported lifetimes of 28.7 s and 24.2 s measured at zero magnetic field? If yes, how exactly was it done? Overall, I was not able to find experimental information about ¹⁵N and ¹³C hyperpolarization. How exactly were the experiments performed? Was the magnetic field in the shield controlled? This information is crucial for data reproducibility.
8. Regarding the same paragraph. It was shown (Barskiy et al., *ChemPhysChem*, 2017) that a significant quenching of ¹³C polarization occurs when quadrupolar nuclei are present in the molecule. Did authors experience lower polarization levels (and/or shorter relaxation times) when using deuterated catalyst analogs? Since deuterium is a quadrupolar nucleus, it can lead to shortening of the relaxation times of the coupled nuclei at zero magnetic field (due to scalar relaxation of the second kind). Please, comment.

Overall, it is a good study which will be of significant interest to the parahydrogen hyperpolarization community. It can influence thinking in the field by providing the example of catalyst optimization strategies to achieve highest NMR signal enhancement.

Reviewer #1 (Remarks to the Author):

This is a rather exhaustive study in which the authors attempt a systematic evaluation and rationalisation of several factors that can potentially affect the levels of hyperpolarisation achievable in SABRE. It will be of interest and of practical use for those working in the field of SABRE hyperpolarisation. However, in my opinion this work is not suitable for publication in Nature Communications for several reasons.

Comment 1. The authors tend to overemphasize the significance of their results. While the word ‘dramatic’ and its derivatives are used a dozen times throughout the manuscript (including the title), an additional signal enhancement of 2-3-fold in NMR, can hardly be called dramatic compared to the enhancement of several orders of magnitude that has been achieved in previous studies. The achieved improvement in SABRE efficiency is certainly useful, but is rather an incremental achievement rather than a major one. From the fundamental point of view, there is not much novelty in this work either. The electronic and steric properties of NHC ligands and their effects on catalyst behaviour have been addressed previously, both in the general catalytic context, and in the context of parahydrogen-based NMR signal enhancement optimisation. The prolongation of polarisation lifetime and an increase in polarisation levels upon deuteration of ligands and substrates has also been reported before. In particular, all this is mentioned in the introduction which also provides relevant references.

Reply: We have reduced the number of times we have used the word dramatic and the title has been changed on suggestion of reviewer 3. We, and the other two reviewers, would disagree with the lack of novelty or significance of the results presented in this manuscript. Whilst we agree that the electronic and steric properties of NHC ligands is well established, there is no rigorous literature analysis for how the nature of these properties influences the SABRE reaction. Indeed, Reviewer #3 notes “*For the first time, the rational design of the SABRE catalysts was driven by the previously developed models and the data obtained confirms the existing theory.*”

We also suggest that it is incorrect to say that the additional signal enhancement is just “2-3-fold”. For example, for ^{15}N polarization under the standard SABRE catalyst

we achieved ca. 20000-fold improvement at 9.4 T (7.1 % polarization), whereas under the conditions and catalyst presented in this manuscript a 125000-fold improvement at 9.4 T. For ^{13}C , these signal enhancements improvements from ca. 2500-fold to 30000-fold. We would also note that cryoprobes, which cost ca. £200,000, provide a 5-8 fold response. Because the response time scales with the square of this number even a 5-fold gain reflects a compression of time by 25 times. Hence we believe the results presented here represent a much more significant breakthrough in the SABRE technique than the reviewer has noted.

From a wider perspective, the paper contains the synthesis of a wide range of novel carbene ligands, and the associated synthetic procedures and characterisation data is included in full in the supporting information. We believe this will impact much wider than merely those working in the SABRE field as we stated in our original letter. The reviewers comment is therefore misplaced.

Comment 2. The phrase “propagating scalar couplings” is most likely slang rather than an actual term and should be avoided throughout the manuscript

Reply: We have changed this phrase throughout the manuscript and now explain it as the spin-spin (or scalar) couplings responsible for polarization transfer.

Comment 3. The manuscript needs careful proofreading as it contains a number of typos and incorrect sentences. The examples are:

where it's modified magnetic properties improve detection...

we show demonstrate the utility of these catalysts...

by replacing then ortho methyl groups...

with electron-poor 21 undergoing slowing dissociation (2.7 s^{-1}) than electron-rich 19 and 20...

the concentrations and relative excess of A was kept constant.

Supporting Information (Fig. SXX).

we rationalize by reduced the spin-spin coupling network within the active catalyst.

Our final aim in this work iss to demonstrate...

A series a samples containing...

and it's fully deuterated isotopologue...

has previously been predicted to be than those commonly reported...

Reply: All above typos have been corrected and other proofreading changes made.

Reviewer #2 (Remarks to the Author):

This manuscript describes a significant advance in the field of SABRE hyperpolarization, and would be of interest to all working in that field of hyperpolarization. The work is aimed at a more rational catalyst design. The study is thoroughly performed with painstaking 138 pages of supporting information containing 122 supporting figures. I have never seen such a long Supporting Information file. This is a compliment, because the contained information will surely be appreciated by those working in this field and developing novel imaging strategies (clearly outlined in the introduction). I highly recommend this manuscript for publication in this top journal. The manuscript is very well written, and I have only a few minor suggestions.

Comment 1. When discussing steric hindrance effect, the authors should cite previous work (R. V. Shchepin, M. L. Truong, T. Theis, A. M. Coffey, F. Shi, K. W. Waddell, W. S. Warren, B. M. Goodson, E. Y. Chekmenev, J. Phys. Chem. Lett. 2015, 6, 1961-1967), where the importance of this effect was realized, and elegantly solved in this paper.

Reply: We have added the suggested reference and other related examples and the following statement.

*The steric bulk of the substrate can also have an effect on the SABRE activity due to binding to the catalyst being inhibited.*²²⁻²³

Comment 2. Most of the record polarization levels reported here are obtained by substrate/catalyst dilution. This may be perceived by some reader as the limitation in the context of biomedical application, where both concentration and polarization are importance factors governing the attainable SNR in MRSI studies. I suggest adding the following text (somewhere): “The experiments performed here reported on the record level of polarization values at relatively low substrate concentration to demonstrate the feasibility of high polarization levels. This is not necessarily a limitation for biomedical applications, because the improved access of substrate/catalyst to parahydrogen may enable such high polarization levels at significantly higher concentrations.” And cite the above-mentioned paper. This can be done in heteronuclear section of the manuscript.

Reply: We have added the suggested discussion of concentration. We felt the conclusions to be a more appropriate placement and now say:

Additionally, whilst the highest levels of polarization reported in this study are obtained at low substrate concentration, this is not necessarily a limit for biomedical applications of SABRE as improved access of the active catalyst to $p\text{-H}_2$ may lead to improved polarization at significantly higher concentrations and even neat liquids.²²

Comment 3. In the sentence: “While metronidazole has already been shown to yield >20% ^{15}N polarization using **1** this result should therefore be amenable to further improvement using the catalysts reported here.⁵⁷” The comparison is not necessarily fair, b/c Ref. 57 reported this polarization level at 150 mM substrate concentration. Moreover, more recent work reported on >34% polarization at 20 mM concentration: B. E. Kidd, J. L. Gesiorski, M. E. Gemeinhardt, R. V. Shechepin, K. V. Kovtunov, I. V. Koptuyug, E. Y. Chekmenev, B. M. Goodson, *J. Phys. Chem. C* 2018, DOI 10.1021/acs.jpcc.1028b05758. This new paper should be additionally cited by the authors.

Reply: We have clarified this statement to include the data that was published whilst our paper was under review. We now say:

*Thus, we predict that for substrates such as the well-studied metronidazole, which has already yielded excellent ^{15}N polarization levels of >34% using **1**, could be amenable to further improvement using the catalysts reported here.⁵⁷⁻⁵⁸ As this is an antibiotic and hypoxia probe, improved catalyst selection could expedite its use in in vivo monitoring.*

Comment 4. The authors may also want to cite the above reference in the next paragraph, where they discuss recently developed tools for translation of SABRE in the realm of biomedical imaging.

Reply: Reference added.

Comment 5. The sentence: “This level of hyperpolarization is commensurate with that created by DNP and successfully used for in vivo study.⁵⁶” should be revised. The choice of DNP reference is subjective. Surely, contrast agents relying on parahydrogen have been demonstrated too. One way to remedy is to revise to ““This level of hyperpolarization is commensurate with that created by the hyperpolarization

techniques, which are already employed for in vivo biomedical imaging.” And cite relevant papers.

Reply: We have made the suggested changes and now say

*This level of hyperpolarization is commensurate with that created by hyperpolarization techniques that have already been successfully used for in vivo biomedical imaging.*⁵⁷⁻⁶¹

Comment 6. The authors discuss relaxation, but surely this is a relaxation in mT or high-field regimes. The relaxation at micro-Tesla magnetic fields is very much different: R. V. Shchepin, L. Jaigirdar, E. Y. Chekmenev, J. Phys. Chem. C 2018, 122, 4984–4996. Since the micro-Tesla fields were employed here too, the authors may want to write a clarifying statement.

Reply: We have clarified the field at which the relaxation rates were measured and note this reference is already used.

Reviewer #3 (Remarks to the Author):

In their manuscript, Rayner et al. tested the performance of more than 30 novel N-heterocyclic carbene (NHC) complexes (catalysts) in the hyperpolarization process Signal Amplification By Reversible Exchange (SABRE). The key characteristics of the catalysts were rationalized (i.e., electronic, steric properties and nuclear spin relaxation) and the most important parameters affecting the hyperpolarization efficiency were identified. For the first time, the rational design of the SABRE catalysts was driven by the previously developed models and the data obtained confirms the existing theory. One of the most important conclusions is that catalyst optimization (fine-tuning) is required for each substrate.

This is an excellent paper which deserves to be published in Nature Communications after minor comments are addressed, see below:

Comment 1. Title of the paper should be modified. Current title does not explain what the paper is going to be about and looks more like a general description of the SABRE process. Moreover, in the last several years Duckett and co-authors published a number of papers having very similar titles (for example, see Rayner et al., Proc. Nat. Acad. Sci., 2017; Lloyd et al., Catal. Sci. Technol., 2014; Adams et al., Science, 2009). I suggest the following title: “Fine-tuning the efficiency of parahydrogen-

induced hyperpolarization experiments by rational catalyst ligand modifications” or something similar which highlights the actual content of the paper.

Reply: We have changed the title to: *Fine-tuning the efficiency of parahydrogen-induced hyperpolarization by rational catalyst design*

Comment 2. Abstract. Avoid the word “non-hydrogenative” when describing SABRE process. While it is generally true that hydrogenation reaction is not required for the observation of the SABRE phenomenon, several studies (Emondts et al. ChemPhysChem, 2018; Barskiy et al., JACS, 2014; Glöggler et al., 2011) have shown the presence of the chemical exchange accompanying the polarization transfer process. Therefore, rigorously, it is incorrect to call SABRE a non-hydrogenative process.

Reply: We have removed the word “non-hydrogenative” as suggested.

Comment 3. Page 3, line 86: remove “show” or “demonstrate”.

Reply: Removed as suggested

Comment 4. Figure 3d. Where is the data for meta-substituents? Was not it measured? If yes, why? It should be specified in the text.

Reply: In the section titled “Reaction of the NHC catalysts with H₂” we explain that the meta substituted catalysts **6** and **7** do not form SABRE active catalysts due to CH activation. Therefore, they do not give ligand dissociation data appropriate for Figure 3d. We have added a clarifying statement at the end of the section to now say

Therefore, catalysts 2, 6, 7 and 22, which did not form active SABRE catalysts, have been removed from the study at this point.

And also to the figure caption which now says:

Fig. 3. a) Tolman Electronic Parameter (TEP) of the NHC ligand, grouped according to change in the *ortho*, *meta*, *para* and *imidazole* substituent. b) Relationship between Ir-C₁ (where C₁ is the carbene centre) bond length and TEP in the corresponding [IrCl(COD)(NHC)]. c) Buried Volume (%V_{bur}) value for these catalysts. d) Rate constants for dissociation of A_{equ} (s⁻¹) in [Ir(H)₂(NHC)(A)₃]Cl (catalysts **2**, **6**, **7** and **22** are omitted due to no active SABRE catalyst formation on exposure to H₂)

Comment 5. Page 8. Lines 162-166. Formation of a new complex showing the hydride resonance sounds interesting. Could authors provide more details about the

proposed structure (e.g., by putting it in supporting information)? Why does not precatalyst **22** form the active catalyst form? It is only different from other catalysts by the subtle change of the ligand. What is the authors' explanation? More discussion is necessary even if it is just a speculation.

Reply: The formation of complexes as the result of C-H activation of phenyl containing NHC ligands is well established. We have stated the following to explain this possibility and given appropriate references.

*For example, **2** lead to a new hydride containing complex that yields a $\delta_H -12.13$ resonance and does not undergo $p-H_2$ addition. This is attributed to the formation of a C-H bond activation product based on the reactivity of related phenyl substituted NHC derivatives.⁴⁸⁻⁵⁰*

The route of decomposition of catalyst **22** is unknown. We agree that the changing of the substituents on the backbone to a bromine seems subtle, however it is likely that the opportunity for oxidative addition into the new C-Br bond (over C-Cl or C-H) is vastly increased.

Comment 6. Page 9, line 202: Fig. SXX – please specify.

Reply: This has been changed to Fig. S15.

Comment 7. Page 14, lines 349-351. Are the reported lifetimes of 28.7 s and 24.2 s measured at zero magnetic field? If yes, how exactly was it done? Overall, I was not able to find experimental information about ¹⁵N and ¹³C hyperpolarization. How exactly were the experiments performed? Was the magnetic field in the shield controlled? This information is crucial for data reproducibility.

Reply: This lifetime was measured at 9.4 T and has been clarified in the text where we now say:

*This change is reflected in the ~20% extension in magnetic state lifetime at 9.4 T which is now 28.7 ± 0.8 s rather than 24.2 ± 0.4 s for systems based on **d₂₂-21** and **1** respectively.*

For ¹³C, the polarization transfer was conducted at ≈ 0.5 G and is stated in the text. We believe transfer is likely to occur via the ¹H in the substrate studied. For ¹⁵N we state in the text that polarization transfer was conducted inside a μ -metal shield (ca. 350-

fold shielding). The magnetic field in the shield was not controlled. We have also added further experimental detail into the supplementary information as follows:

5.1 Sample preparation and SABRE method

A 5 mm J. Young's tap NMR tube containing a 5 mM (unless otherwise stated) solution of [IrCl(COD)(NHC)] and substrate (4 eq.) in methanol-d₄ (0.6 mL) was degassed prior to the introduction of p-H₂ (3 bar unless otherwise stated). Samples were then shaken for 10 s in the specified polarization transfer field before being rapidly transported into the magnet for subsequent interrogation by NMR spectroscopy. For ¹H, the typical polarization transfer field was 65 G, for ¹³C it was 0.5 G and for ¹⁵N the sample was shaken inside a μ-metal shield with ca. 350-fold shielding.

Comment 8. Regarding the same paragraph. It was shown (Barskiy et al., ChemPhysChem, 2017) that a significant quenching of ¹³C polarization occurs when quadrupolar nuclei are present in the molecule. Did authors experience lower polarization levels (and/or shorter relaxation times) when using deuterated catalyst analogs? Since deuterium is a quadrupolar nucleus, it can lead to shortening of the relaxation times of the coupled nuclei at zero magnetic field (due to scalar relaxation of the second kind). Please, comment.

Reply: We did not see any reduction in polarization levels when using the deuterated catalyst analogues. This may indicate that the coupling to quadrupolar nuclei in the catalyst has negligible effect when compared to the couplings into the substrate molecule.

Overall, it is a good study which will be of significant interest to the parahydrogen hyperpolarization community. It can influence thinking in the field by providing the example of catalyst optimization strategies to achieve highest NMR signal enhancement.

We hope that our detailed reply full answers the questions of the reviewers and we can now proceed to publication.

Reviewer #3 (Remarks to the Author):

Authors' reply is satisfactory. A revised version of the paper and supporting information are ready for publication.